# Switching to an Infliximab Biosimilar Was Safe and Effective in Dutch Sarcoidosis Patients

**DOI:** 10.3390/cells10020441

**Published:** 2021-02-19

**Authors:** Bas J. M. Peters, Anish Bhatoe, Adriane D. M. Vorselaars, Marcel Veltkamp

**Affiliations:** 1Department of Clinical Pharmacy, St. Antonius Hospital, 3430 EM Nieuwegein, The Netherlands; anishbhatoe@gmail.com; 2ILD Center of Excellence, St. Antonius Hospital, 3430 EM Nieuwegein, The Netherlands; a.vorselaars@antoniusziekenhuis.nl (A.D.M.V.); m.veltkamp@antoniusziekenhuis.nl (M.V.); 3Division of Heart and Lungs, Department of Pulmonology, University Medical Centre, 3508 GA Utrecht, The Netherlands

**Keywords:** biosimilar, infliximab, sarcoidosis, TNF-α inhibitor

## Abstract

The effect of switching from originator infliximab to biosimilar infliximab in patients with sarcoidosis is unknown. The objective of this study is to investigate the effect of switching from Remicade^®^ or Inflectra^®^ to Flixabi^®^ in patients with severe refractory sarcoidosis. This single center retrospective cohort study was performed at St Antonius Hospital Nieuwegein, The Netherlands. All patients diagnosed with severe refractory sarcoidosis receiving Remicade^®^ or Inflectra^®^ switched to Flixabi^®^. The primary outcome was infliximab discontinuation within 6 months of switching. Secondary endpoints included adverse events and loss of clinical, functional, or inflammatory response. Out of 86 patients who switched to Flixabi^®^, 79 patients had complete data. None of the 79 patients discontinued infliximab during the first 6 months after switching. Five patients reported an adverse event related to Flixabi^®^ treatment. We found no change from baseline in FVC, FEV1, DLCOc, 6MWT, and infliximab trough levels 26 weeks after switching. An improvement in physical functioning of 7.3 ± 13.4 points (*p* = 0.002) with RAND/SF36 and in biomarker sIL-2R (−475.58 ± 1452.39; *p* = 0.005) was observed. Switching from originator infliximab Remicade^®^ or biosimilar infliximab Inflectra^®^ to biosimilar infliximab Flixabi^®^ did not result in treatment discontinuation or loss of clinical/functional/inflammatory remission.

## 1. Introduction

Sarcoidosis is an immune-mediated inflammatory disease, characterized by systemic granulomatosis, which affects many organs and often has an impact on the quality of life [1]. In the Netherlands, the estimated annual incidence of sarcoidosis is 20 cases per 100,000, with a prevalence of 50 cases per 100,000 [2]. Patients with a severe and/or progressive course of the disease require medical intervention, aiming for a reduction of symptoms and prevention of organ damage. Initially, patients are treated with systemic glucocorticoids and/or disease-modifying antirheumatic drugs (DMARDs). If the disease progresses or patients experience toxic effects of these therapies, targeted TNF-α inhibition can be considered [3].

In multiple studies, the off-label use of infliximab in sarcoidosis has been demonstrated to result in clinical improvement [4,5,6,7,8,9]. It is postulated that cytokine TNF-α has an accelerating effect on the inflammatory process in sarcoidosis because of its role in granuloma formation [10].

In 2015, a position paper guiding the use of infliximab biosimilars in sarcoidosis patients was published by the Dutch Association of Pulmonologists (NVALT) [11]. In this paper, it was recommended to treat new patients with an indication for TNF-α blockers with the biosimilar Inflectra^®^ instead of the originator Remicade^®^ to reduce healthcare costs. Subsequently, an observational study was conducted in our hospital in sarcoidosis patients requiring infliximab treatment starting with the biosimilar Inflectra^®^. The results of this study showed that the therapeutic effect and the safety profile between the biosimilar and original biological are comparable [12].

In the meantime, several switch studies have been conducted demonstrating the safety of switching from originator infliximab to biosimilar infliximab in patients receiving anti-TNF treatment [13,14,15,16,17,18,19]. This was to be expected given the extensive product requirements manufacturers of biosimilars have to fulfill before receiving a market authorization. However, since these trials included patients with either arthritis [14,16,17], ankylosing spondylitis [13], and/or inflammatory bowel disease [16,18,19], data on actively switching to biosimilar infliximab in sarcoidosis are lacking.

The primary objective of this study is to determine safety and efficacy after either switching from an infliximab originator to a biosimilar or within the group of infliximab biosimilars in Dutch patients with sarcoidosis.

## 2. Materials and Methods

This single center retrospective cohort study was performed at St. Antonius Hospital Nieuwegein, The Netherlands, a national tertiary referral center for sarcoidosis patients. In line with guidance on switching biologicals provided by the Dutch Federation of Medical Specialists (FMS) and Dutch Medicines Evaluation Board (CBG), all patients (aged 18 years or older) diagnosed with severe refractory sarcoidosis and receiving Remicade^®^ or biosimilar Inflectra^®^ maintenance treatment in the time window 2018–2019 were switched to Flixabi^®^ as part of routine clinical care.

Study data were collected by medical chart review and managed using REDCap electronic data capture tools hosted at St. Antonius Hospital, Nieuwegein. After completion of the database, the database was exported with a unique identifier (studyID) which was different from the original patient identifier. Laboratory outcomes, clinical outcomes, and quality of life outcomes at 26 weeks after the switch were compared with these outcomes at the time of the switch. The study was approved by the Institutional Review Board (or Ethics Committee) of St. Antonius Hospital (protocol number Z20.137, date 8 December 2020).

### 2.1. Outcomes and Study Parameters

The primary outcome of the study was infliximab discontinuation within 6 months of switching. Secondary endpoints included adverse events, the formation of anti-drug antibodies (ADAs), and loss of clinical, functional, or inflammatory response. Clinical status was assessed by the RAND 36 item of physical functioning (as reported by Vorselaars et al.) [9]. A maximum score of 100 in the RAND/SF36 questionnaire indicates that there is no disability in the measured domain, which is physical functioning in this case, and consequently, a score of zero indicates that there is a maximum disability. Functional status was determined for patients with a pulmonary treatment indication and included Forced vital capacity (FVC), forced expiratory volume in 1s (FEV1), diffusing capacity of the lung for carbon monoxide (DLCOc), and the 6-min walking distance. Inflammatory status was assessed by measurement of angiotensin-converting enzyme (ACE) and soluble Interleukin-2 receptor (sIL-2R). ADAs were measured in patients when deemed relevant by the treating physician in case of loss of response, adverse reactions, or low infliximab through levels. Study parameters include gender, date of birth, date of diagnosis, age at switch date, ethnicity, Scadding stage, main treatment indication, concomitant immunosuppressive drugs, and extra-pulmonary manifestations.

### 2.2. Statistical Analysis 

SPSS Statistics (version 24.0) was used for the statistical analyses. Continuous data are expressed as mean ± SD or as the median (interquartile range) where appropriate. Categorical data are expressed as frequencies (%). Categorical data are analyzed by Chi-square and continuous data by a paired *t*-test.

## 3. Results

A total of 86 patients were identified who were receiving maintenance treatment with Remicade^®^ or Inflectra^®^ and were switched to Flixabi^®^. Seven patients were excluded due to immature data at the time of data collection. None of the patients switched infliximab brands before. Patients had an average age of 51.8 years (SD 11.1), had a disease duration of almost 9.8 years (SD 6.5), and were treated with infliximab (Remicade^®^ or biosimilar in-fliximab Inflectra^®^) for a mean duration of 4.4 years (SD 2.7). All patients were treated at a dose of 5 mg/kg except for one patient who was treated with 7.5 mg/kg. The dose did not change in the time after switching for any patient. Treatment intervals were the same for 60 (76%) patients during follow-up, whereas the interval was prolonged with one and two weeks for respectively eight (10%) and 10 (13%) patients. Only one patient was dosed at a one-week shorter interval after switching. Half of the patients had a pulmonary treatment indication. A complete overview of the baseline characteristics is presented in Table 1.

### 3.1. Primary Endpoint

None of the 79 patients included in this study had to discontinue infliximab during the first 6 months after active switching from infliximab originator to a biosimilar or within the group of infliximab biosimilars.

### 3.2. Secondary Endpoints

#### 3.2.1. Safety

Out of thirteen patients who reported adverse events related to infliximab prior to switching, three patients also reported an adverse event related to Flixabi^®^ treatment. In total, five patients reported an adverse event related to Flixabi^®^ treatment. No major adverse events were reported. None of the patients experienced infusion reactions. Two patients experienced general malaise and three patients experienced arthralgia or muscle strains who were switched back to Remicade^®^. Two patients were switched back after 6, 5, and 9 months and reported an increase in quality of life after the switch back to Remicade^®^. Infliximab trough levels did not vary significantly over time regardless of Remicade^®^ or Flixabi^®^ treatment for both patients. An overview of the adverse events related to infliximab presented in Table 2.

#### 3.2.2. Infliximab Trough Levels and Antidrug Antibodies

Infliximab trough levels before and after the switch did not significantly change when comparing the last trough level before the switch to both the first subsequent trough level (Table 3). In addition, no difference was observed when infliximab trough levels, prior to and after switching, were compared of patients treated with either Remicade^®^ or Inflectra^®^. ADA levels were determined in seven patients but no detectable ADA levels were observed. The majority of patients had the same treatment interval at the time of the switch and 26 weeks after the switch (*n* = 64). The treatment interval was prolonged in 14 patients and shortened for one patient at the moment of the switch.

#### 3.2.3. Clinical Outcomes

##### Pulmonary Function

Data on pulmonary functioning was available in 41 patients. No differences were found between baseline pulmonary function before and after the switch (Table 4). Also for pulmonary sarcoidosis patients only, no differences were found in the change in pulmonary function after the switch to Flixabi^®^ (data not shown). Finally, after switching an improvement of 7 ± 13 points was found for physical functioning as measured using the RAND/SF36 (*p* = 0.002).

##### Inflammatory Response

The biochemical disease activity parameter ACE did not change after the switch whereas sIL-2R was found to be significantly lower after the switch (−475 ± 1452; *p* = 0.005) (Table 3). Concomitant use of immunosuppressive therapy did not change (data not shown).

## 4. Discussion

This study described the effect of switching from either infliximab originator Remicade^®^ or biosimilar Inflectra^®^ (CT-P13) to infliximab biosimilar Flixabi^®^ in patients with severe refractory sarcoidosis. None of the sarcoidosis patients included in this study had to discontinue infliximab during the first 6 months after active switching to Flixabi^®^. We found that switching was safe and resulted in an overall persisting clinical benefit as reflected by physical functioning, lung function parameters, and inflammatory disease activity before and after the switch. This is in line with previous studies, both randomized clinical trials [13,14,15] and real-world [17,18], on switching in other diseases than sarcoidosis. Two studies on switching to specifically Flixabi^®^ in inflammatory bowel disease and rheumatoid arthritis patients also showed comparable efficacy, safety, and immunogenicity profiles for switching and non-switching study arms [16,19]. Furthermore, we found no evidence of ADA formation based on adequate trough levels throughout 26 weeks of treatment. Of note, in contrast to the previous switch studies, our study did not only include patients who were switched from the originator infliximab, but it also included patients who were switched from the widely used biosimilar Inflectra^®^ (CT-P13).

In our study, no patient discontinued Flixabi^®^ during the first 6 months after the switch. This is in line with a recent study by Bronswijk et al. which reported a 6-month discontinuation rate in IBD after switching of 4% [20]. Other switching studies have reported higher discontinuation rates (12.2–28.2%) than would be expected based on rates observed in blinded switching studies [21]. Interestingly, it has been postulated that the majority of these treatment failures result from a “nocebo effect”, defined as patients’ negative expectations toward the therapy change highlighting the importance of patient education and physician engagement in the process of non-medical switches [21,22]. Furthermore, 6% of patients had adverse events related to Flixabi^®^. Reported side effects after the switch were general malaise, fatigue, arthralgia, and myalgia. Nevertheless, two patients did switch back to Remicade^®^ because of a decline in quality of life during treatment with Flixabi^®^. Given the subjective nature of the treatment indication for these two patients, it can be debated whether the decline in quality of life was a result of a failure of the pharmacological origin or of the reluctance of switching in the context of infliximab as a last-resort treatment for sarcoidosis. 

Recently, Kidd et al. reported a relapse of severe neurosarcoidosis which was suggested to be caused by the switch from originator infliximab to biosimilar infliximab [23]. Another study by Riller et al. described both initiation and switching of infliximab treatment in 20 neurosarcoidosis patients [24]. Although six patients relapsed during biosimilar treatment, relapse rates and time-to-relapse did not differ between the infliximab originator previously received and biosimilar treatment groups. In our study, there was no relapse observed in seven neurosarcoidosis patients during the 6 months follow-up. In addition, Xue et al. conducted a prospective observational monitoring study of 48 rare immune-mediated inflammatory diseases patients (among which 17 with sarcoidosis) that were switched from Remicade^®^ to Remsima^®^ [25]. No significant differences were observed for patient-reported outcomes, physician perception disease activity, and laboratory parameters after switching from Remicade^®^ to Remsima^®^. However, the study did report therapy failure or exacerbation of neurological symptoms in a substantial number of patients who were diagnosed with granuloma, including (neuro)sarcoidosis. In contrast to Xue et al., we did not observe a large number of patients to have treatment failure/exacerbation during our 6 months versus their 2 years [25] follow-up.

Interestingly, we found an improvement in both physical functioning with the RAND/SF36 questionnaire and biomarker sIL-2R. One can speculate on the origin of these findings. Glycosylation profiles and in vitro biological activity show some variation between infliximab brands [26] and monoclonal antibody batches [27] although the antigenic properties are interchangeable between different brands [28]. However, based on the majority of clinical studies these variations appear to be clinically irrelevant. For sarcoidosis, recent papers demonstrate that the mTORc pathway in combination with autophagy appears to play an important role in the pathogenesis of sarcoidosis [29]. Variations between the different infliximab brands (for example the glycosylation status) could theoretically modulate these regulation processes and consequently biological and functional markers such as sIL-2R and physical functioning. However, based on our data we cannot make any statement about a possible different effect on these pathways by reference product or biosimilar. In fact, the improvement of two outcomes may be a result of chance or an ongoing infliximab treatment effect. The further improvement in biomarkers and quality of life seen after switching in our patients with sarcoidosis support the general view that switching of infliximab from originator product within the class of biosimilars is safe. Most importantly, our study was performed to show non-inferiority.

Our study has several strengths and limitations. First, this is the largest study so far that describes switching infliximab within a population of severe refractory sarcoidosis patients. Second, although the retrospective observational nature of our study, there was limited missing data of a comprehensive set of primary and secondary outcomes to assess the ongoing response of infliximab treatment after switching compared to before switching. Clinical outcomes, as well as biochemical and pharmacokinetic parameters of patients switching infliximab, were evaluated in a real-world setting. One limitation of our study may be considered the duration of follow-up. Patients could have experienced loss of response beyond the minimum duration of 26 weeks follow-up.

To conclude, building upon existing evidence, we showed switching to and between infliximab biosimilars to be effective and safe also in sarcoidosis patients which can potentially further reduce healthcare costs [30]. Switching from either originator infliximab Remicade^®^ or biosimilar Inflectra^®^ to biosimilar infliximab Flixabi^®^ did not result in treatment discontinuation or loss of clinical, functional, or inflammatory remission.

## Figures and Tables

**Table 1 cells-10-00441-t001:** Baseline characteristics of sarcoidosis patients at the time of the switch from Remicade^®^ or Inflectra^®^ to Flixabi^®^.

	Switch Group (*n* = 79)	%
Sex (female)	33	42
Ethnicity (non-white)	10	13
Age; mean years (SD)	51.9 (11.2)	
Disease duration; mean years (SD)	9.8 (6.5)	
Duration of ongoing infliximab treatment; mean years (SD)	4.4 (2.7)	
Weight; mean kg (SD)	93.9 (21.0)	
Main treatment indication		
Pulmonary	41	52
Cardiac	13	17
Small Fiber Neuropathy	7	9
Central Nervous System	7	9
Hypercalcemia	5	6
Other	6	8
Scadding stage		
0	10	13
1	20	25
2	27	34
3	6	8
4	16	20
Infliximab brand before the switch		
Remicade^®^	24	30
Inflectra^®^	55	70
Treatment interval (weeks)		
4	55	70
5	4	5
6	12	15
7	2	3
8	6	8

**Table 2 cells-10-00441-t002:** Overview of adverse events related to infliximab therapy before switching (Remicade^®^ or Inflectra^®^) and after switching (Flixabi^®^).

	Before Switching (n)	%	After Switching (n)	%
Infusion reaction	1	1.3		
General malaise, fatigue	1	1.3	2	2.5
Arthralgia or myalgia	3	3.8	3	3.8
Cutaneous reaction	4	5.1		
Blurred vision	1	1.3		
Headache	1	1.3		
Forgetting/amnesia	1	1.3		
Obstipation	1	1.3		
None	66	83.5	74	93.7

**Table 3 cells-10-00441-t003:** Infliximab trough levels, disease activity, and health-related quality of life (physical functioning) at baseline and 26 weeks after switching to Flixabi^®^ treatment; Mean (SD).

	Baseline	After 26 Weeks Flixabi ^®^	*p*-Value
*Disease activity/severity measurements*	N = 79		
ACE (U/l) (SD)	39 (20)	40 (20)	0.27
sIL-2R (pg/mL) (SD)	3169 (1985)	2694 (1464)	0.005
*Infliximab trough levels*	N = 58		
Trough level (mg/L) * (SD)	19.5 ± 15.1	19.6 (16.2)	0.82
*Health-related quality of life*	N = 39		
RAND/SF36 score on physical functioning (SD)	42 (22)	49 (24)	0.002

ACE = Angiotensin-converting enzyme; sIL-2R = soluble Interleukin 2-receptor; * Last infliximab trough level before switching and first trough level after switching are compared.

**Table 4 cells-10-00441-t004:** Pulmonary function at baseline and 26 weeks after switching to Flixabi^®^ for patients with a pulmonary treatment indication; Mean (SD).

	**Baseline**	**After 26 Weeks Flixabi^®^**	***p*** **-Value**
	N = 41		
**FVC, % predicted (SD)**	81.9 (19.1)	82.3 (18.6)	0.76
**FEV1, % predicted (SD)**	67.3 (19.8)	66.6 (19.5)	0.52
**DLCOc, % predicted (SD)**	64.9 (14.4)	65.3 (12.9)	0.82
**6MWT, % predicted (SD)**	81.0 (21.9)	81.0 (23.0)	1.00

FVC = Forced vital capacity; FEV1 = Forced expiratory volume in 1 s; DLCOc = Diffusing capacity for carbon monoxide; 6MTW = six-minutes walking test.

## Data Availability

The data presented in this study are available on request from the corresponding author. The data are not publicly available due to privacy.

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
