# Peer review of "Switching to an Infliximab Biosimilar Was Safe and Effective in Dutch Sarcoidosis Patients"

_cells, 2021, doi:10.3390/cells10020441_

Round 1

Reviewer 1 Report

This is a well-written manuscript with a clear message, it is easy to understand and the results are of high importance for clinicians working with sarcoidosis patients.

I have some questions and remarks:

  1. It is an observational study, I agree, but is that a reason for not getting informed consent from the patients? Is that really OK according to Dutch law?
  2. RAND/SF36 was used for measurement of physical functioning. The Fatigue Assessment Scale is validated for measurment of fatigue in sarcoidosis and as fatigue is  a symptom that bothers many patients with sarcoidosis I wonder why that scale wasn´t used? Also, as sarcoidosis affect multiple dimensions of the patient´s lives it would have been interested if the authors had used King´s Sarcoidosis Questionnaire. Why wasn´t that used?
  3. Only 7 patients were tested for ADA. However, in conclusions it is stated that Flixabi did not result in formation of ADA. I can admit that it is not so probable that ADA were formed as the therapy seemed to function well. Still, the presence of ADA was only investigated in 7 patients. Therefore, I suggest this part of the conclusion is removed.
  4. I could not find information about mean dose (mg/kg) and infusion intervals for the given drugs. I would like to have that information.
  5. About safety, 3.2.1. I am not sure I understand fully what is meant. I understand the text that only 5 patients reported AE related to Flixabi and that all these 5 also had experienced AE during treatment with infliximab prior to switching. Is that correctly understood? If so, I think that information is important and should be stressed. If I did not understand this correctly, I think this part must be re-written.
  6. I find the discussion appropriate and it iseasy to follow the authors thoughts. However, there are some strange typing mistakes, e.g. prod-uct, de-scribes, num-ber. 

Reviewer 2 Report

Even if the efficiency of the two products was not significantly different, the authors highlighted two positive variations, one for the interleukin 2 receptor as a biological marker of the therapeutic response, the second for the RAND/SF36 physical test.

My major comments:

  1. The work presented in this manuscript is retrospective and share the weakness of this type of study. Nevertheless, it’s the first one comparing these two antibodies in the field of sarcoidosis. This new monoclonal antibody has been already compared with the main reference product, INFLIXIMAB (REMICADE©) in various types of arthritis, ankylosing spondylitis and inflammatory bowel diseases and has shown similar efficiency with a decreased cost. We believe that this work must be cited and discussed by the authors. In addition, the antigenic properties of this new monoclonal antibody are superimposable to the reference Ab.

Reference:

Magro F, Rocha C, Vieira AI, Sousa HT, Rosa I, Lopes S, Carvalho J, Dias CC, Afonso J. The performance of Remicade®-optimized quantification assays in the assessment of Flixabi® levels. Therap  Adv Gastroenterol. 2018 Sep 23; 11: 1756284818796956.

  1. It would have been interesting to discuss the glycosylation states of the reference and new monoclonal Ab’s, as detailed in the following reference :

Lee C, Jeong M, Lee JJ, Seo S, Cho SC, Zhang W, Jaquez O. Glycosylation profile and biological activity of Remicade® compared with Flixabi® and Remsima®. MAbs. 2017 Aug / Sep; 9 (6): 968-977.

In fact, the distinct glycosylation states have no impact on the activity of the antibodies but may have some impact during the process of degradation by the proteasome. We might consider that constitutional difference between individuals due to genetic variations may have some consequences of the rate of degradation and by consequence, modify the biological parameters studied such as IL2R marker and/or the physical function assessed by RAND/SF36. 

These differences in glycosylation do not seem to have an impact on the activity of the antibody but it is possible that a different degradation by the proteasome, linked to genetic variations inherent in sarcoidosis, could modify certain biological aspects mentioned in the item such as the level of the IL2R marker or the physical function assessed with RAND / SF36.

  1. The improvement observed for these two markers merits some assumptions. A chronic fatigue syndrome has been often reported in sarcoidosis and explains probably the differences in exercise tolerance. Modulation of TNF alpha has main molecular and physiological impacts and mainly by regulating the mTOR hub and all related metabolic pathways, such as autophagy which has been shown to be putatively affected in sarcoidosis. Some metabolic variations between the different antibodies (for ex. the glycosylation status) may modulate these regulation processes and consequently modulate the mTOR-related autophagy level and the downstream biological and functional markers. A significant number of works has been published on this topic and the TNF alpha / anti TNF alpha / autophagy relationships. It would be important to include this topic in the discussion of the data and observed differences in the two types of markers.

  1. A last point concerns the cost benefit related to this new molecule and the following reference could be included

Aladul MI, Fitzpatrick RW, Chapman SR. The effect of new biosimilars in rheumatology and gastroenterology specialties on UK healthcare budgets: Results of a budget impact analysis. Res Social Adm Pharm. 2019 Mar; 15 (3): 310-317.

In summary, we consider that this work is an interesting issue in the management of severe sarcoidosis and relevant for publication after taking into account the previously detailed comments

Reviewer 3 Report

A study of generic formulation of biosimilar infliximab in patients with sarcoidosis.

-the current PDF has many broken words that were not made whole after PDF process

-further discussion of the salutatory effect of generic formulation in the discussion section is indicated
